# A Teacher-Student Perspective on the Dynamics of Learning Near the Optimal Point

## Abstract

Near an optimal learning point of a neural network, the learning performance of gradient descent dynamics is dictated by the Hessian matrix of the loss function with respect to the network parameters. We characterize the Hessian eigenspectrum for some classes of teacher-student problems, when the teacher and student networks have matching weights, showing that the smaller eigenvalues of the Hessian determine long-time learning performance. For linear networks, we analytically establish that for large networks the spectrum asymptotically follows a convolution of a scaled chi-square distribution with a scaled Marchenko-Pastur distribution. We numerically analyse the Hessian spectrum for polynomial and other non-linear networks. Furthermore, we show that the rank of the Hessian matrix can be seen as an effective number of parameters for networks using polynomial activation functions. For a generic non-linear activation function, such as the error function, we empirically observe that the Hessian matrix is always full rank.

## 1 Introduction

Neural networks have achieved tremendous success in the last decade. Their practical usefulness and technological potential is now undeniable. However, there is an enormous gap between our current theoretical understanding and the state-of-the-art techniques used in recent applications.

Among the many unsolved theoretical puzzles, understanding the generalization and robustness of trained models is particularly important since modern neural networks often work with a number of parameters vastly larger than the amount of available training data. The unexpected effectiveness of stochastic gradient descent as a training method, for high-dimensional and non-convex learning tasks, is also one of the many empirical observations that still lack a consensual explanation.

In this context, understanding the universal properties of the dynamics of learning in high-dimensional neural networks, although particularly challenging, has the potential of unveiling some of the magic behind their remarkable practical effectiveness. In this work, we study learning under gradient descent dynamics in simple, yet nontrivial examples, in an attempt to characterize the effectiveness of the learning process in terms of the features of the network, for a regression task with a single neuron output.

We make two important simplifying assumptions:

1. We focus on a *teacher-student setup* (Zhang et al., 2019; Goldt et al., 2020; Safran et al., 2021; Akiyama & Suzuki, 2023). Here, a neural network, called the *student*, is tasked with learning another fixed neural network, the *teacher*, through its outputs. The learning problem is completely determined by the student and teacher architectures. Thus, empirical claims, which are typically difficult to validate in less academic setups, can be effectively verified using numerical tools and sometimes even analytically.

2. We assume that the network's initialization is sufficiently close to an optimal point. In this case, the Hessian matrix of the loss function with respect to the network's parameters fully characterizes the loss landscape in quadratic order. It is worth noting that previous analyses of the Hessian have

been used to evaluate the flatness and overall curvature of the loss function and its rank used as an effective measure of the number of the network's parameters (Maddox et al., 2020; Singh et al., 2021).

Under these assumptions, we are seeking answers to the following questions:

- How does the evolution of the loss function for a student initialized near the optimal point depend on the characteristics (number of parameters, activation functions) of the network?

- Under which conditions can the Hessian rank at the optimal point be interpreted as an *effective number of parameters*?

All the code supporting this research is available in the provided supplementary material.

**Related work**  In 2021, Singh et al. (2021) published an analytical study of the Hessian rank of deep linear networks, providing tight upper bounds. For non-linear networks, they found that the linear formulas were still empirically valid in determining the numerical Hessian rank. That study came at a time where empirical investigations into the eigenspectrum of the Hessian matrix were being performed (Sagun et al., 2017; 2018). In our work, we determine the Hessian rank at the optimal point, in the teacher-student setup, and we propose using the Hessian rank as a measure of the effective number of parameters. We also provide upper bounds for the Hessian rank at the optimal point for polynomial networks.

Still related to the Hessian matrix, Liao & Mahoney (2021) in 2021 studied the Hessian eigenspectra of nonlinear neural networks, with the objective of understanding the effect of some simplifying assumptions made in the literature to turn the Hessian spectral analysis tractable. Those authors performed a theoretical analysis using random matrix theory that did not make such simplifying assumptions. They found that the Hessian eigenspectra for a broad category of nonlinear models can have different behaviors. They can exhibit either single or multiple bulks of eigenvalues; they can have isolated eigenvalues away from the bulk, and even distributions with bounded and unbounded support. In our work, we shall see that some of the eigenspectra we find also exhibit these very different kinds of spectral behavior.

The teacher-student setup as an investigative tool had a revival in recent work by Goldt et al. (2020), where they studied the dynamics of online stochastic gradient descent for a two-layer teacher-student setup. They focused on the effect of overparameterization of the student network with respect to the teacher network and found that, when training both layers, the generalization error either stayed constant or decreased with student size, depending on the specific activation function chosen for both networks. This result provided a rigorous foundation for a series of earlier papers that studied the teacher-student setup in soft committee machines (Biehl & Schwarze, 1995; Saad & Solla, 1995a;b). Although their object of study is different from ours, we share some of the underlying assumptions, namely, the input data distribution in the teacher-student setup. However, in 2020, Arjevani & Field (2020), used the teacher-student setup to theoretically compute the Hessian eigenvalues at the global minimum and other local minima, for a restricted class of shallow ReLU networks, becoming one of the first theoretical works to find a skewed Hessian eigenspectrum, where most eigenvalues concentrate around zero.

Finally, one concept that is connected to the Hessian and has also been an important tool in theoretical Machine Learning is the Fisher information matrix (FIM), which plays the role of a Riemannian metric in parameter space (Karakida et al., 2020). In 2018, Pennington & Worah (2018) studied the eigenspectrum of the Fisher information matrix in the limit of infinite network input, output and hidden dimensions, using tools from random matrix theory. Other works, such as Karakida et al. (2020; 2021), explored the eigenvalue statistics of the Fisher information matrix, finding that most eigenvalues are concentrated close to zero, with a few rarer but much larger eigenvalues. As shown in Appendix D, the Hessian matrix at the global optimum is equal to the expected FIM, making our analysis of the Hessian rank deeply connected to these works that study the FIM.

**Paper structure**  In the next subsection, we formally describe the teacher-student setup as well as the notation used in the rest of this work. In Section 2, we derive the associated gradient and Hessian equations

as well as the learning dynamics near the optimal point. In Section 3, we answer our two main questions for linear networks, where we are able to fully characterize the probability distribution for the eigenvalues of the Hessian matrix. In Sections 4 and 5, the same analysis is performed for networks with polynomial and another non-linear activation function, the error function. Finally, in Section 6, we summarize our work and discuss possible future work.

## 1.1 Teacher-Student Setup

We use a teacher-student setup where a neural network, called the student, has to learn a function represented by yet another neural network, called the teacher. The teacher network is a fixed neural network which is randomly initialized and is not trained, serving only to create a learning problem that we can tune to alter its complexity.

We can control several parameters in both the student and the teacher networks, like the number of layers, the activation function or whether there are any biases in the linear transformations at each layer. Figure 1 depicts the neural networks under study. However, to be able to study the behavior of the student network at the optimal point, we assume that the parameters that define the architectures of both the teacher and student are the same. Doing so, enables us to always know one optimal solution where the student is capable of reproducing the output of the teacher, which is when all the weights and biases of the student coincide with those of the teacher network.

To further simplify the analysis we make the following architectural choices:

- We work with two-layer neural networks, with one hidden layer with a given activation function, and one linear output layer (no activation function).

- We do not use biases, so that at each layer the pre-activation vector is given by a linear map.

- The output is always a single real number.

- The weights of the teacher network are taken from a normal distribution centered at zero with variance $\frac{1}{N}$, where $N$ is the size of the previous layer.

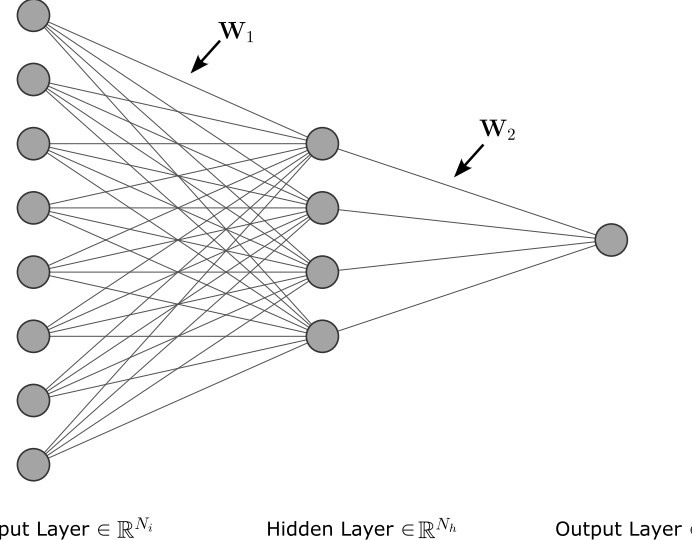

Input Layer $\in \mathbb{R}^{N_i}$          Hidden Layer $\in \mathbb{R}^{N_h}$          Output Layer $\in \mathbb{R}$

Figure 1: Depiction of the architecture of neural networks considered.

**Notation used throughout the article**  We denote by $N_i$ the size of the input layer so that $\boldsymbol{x}$ is a vector in $\mathbb{R}^{N_i}$ referred to as input. Similarly, we denote by $N_h$ the size of the hidden layer, i.e., the number of

neurons in the hidden layer. Matrix $\boldsymbol{W}_1$, which is a $N_h \times N_i$ matrix, represents the linear transformation between the input and the hidden layer. Similarly, matrix $\boldsymbol{W}_2$ is the linear transformation between the hidden layer and the output: it is a $1 \times N_h$ matrix, which can be seen as a row vector due to our choice of only having a real scalar output.

The output of the student network is represented by $y$. Under the above architectural choices, the output of the student network is given by

$$y = \boldsymbol{W}_2 \; g\left(\boldsymbol{W}_1 \boldsymbol{x}\right) \;,$$

where $g : \mathbb{R} \to \mathbb{R}$ is the activation function that acts on the entries of the pre-activation vector $\boldsymbol{z} = \boldsymbol{W}_1 \boldsymbol{x}$, yielding the hidden vector $\boldsymbol{h}$.

We denote the vector of all the network parameters by $\theta$. Any of the previous quantities with an additional hat above denotes a parameter of the teacher network, e.g. $\theta^\star$ represents the set of parameters of the teacher and $y^\star$ the output of the teacher network.

We also have some usage conventions for indices. The index $i$ is only used to identify a sample in a given batch. As such, when talking about inputs $\boldsymbol{x}$, hidden vectors $\boldsymbol{h}$, or pre-activation values $\boldsymbol{z}$, the first index is the batch index if and only if it is an $i$. Otherwise, we assume that a batch size of one is being used. To further simplify the notation, we represent the $(m, n)$ entry of the weight matrix $\boldsymbol{W}_1$ as $W_{1mn}$ and similarly for $\boldsymbol{W}_2$.

## 2 Hessian and Learning Equations

The loss function used in this study is the mean square error (MSE), which, for a batch size of $N$ input vectors, takes the form

$$\mathcal{L} = \frac{1}{2N} \sum_{i=1}^{N} \left(y_i - y_i^\star\right)^2 \;,$$

where $y_i$ is the output of the student network and $y_i^\star$ is the output of the teacher network.

In the simplest case, where the student is a two-layer neural network without any biases, the Hessian matrix for the derivatives of the loss function with respect to the parameters of the student can be decomposed into three different submatrices, or blocks, as

$$\boldsymbol{H} = \begin{pmatrix} \boldsymbol{A} & \boldsymbol{C} \\ \boldsymbol{C}^T & \boldsymbol{B} \end{pmatrix} \;.$$

Matrix $\boldsymbol{A}$ contains the derivatives with respect to weights of matrix $\boldsymbol{W}_1$. Similarly, $\boldsymbol{B}$ contains the derivatives with respect to weights of matrix $\boldsymbol{W}_2$. Finally, the matrix in the antidiagonal, $\boldsymbol{C}$, contains all the cross-derivatives with respect to the two different layers.

Taking the derivative of the loss function with respect to the parameters leads to

$$\frac{\partial \mathcal{L}}{\partial W_{1mn}} = \frac{1}{N} \sum_{i=1}^{n} \left(y_i - y_i^\star\right) W_{2m} \; g'(z_{im}) x_{in} \;, \tag{1}$$

$$\frac{\partial \mathcal{L}}{\partial W_{2k}} = \frac{1}{N} \sum_{i=1}^{N} \left(y_i - y_i^\star\right) h_{ik} \;. \tag{2}$$

Note that, as we will work with the above equations at the optimal point, the two terms that depend on $y_i - y_i^\star$ are always zero. Under the notation and definitions of Singh et al. (2021), this corresponds to only studying what the authors call *outer product Hessian*, since the *functional Hessian*, which depends on the term $y_i - y_i^\star$, is zero. It is also worth noting that the *outer product Hessian* shares the same nonzero eigenspectrum as the Neural Tangent Kernel (NTK) (Jacot et al., 2018) and coincides, at the optimal point, with the Fisher information matrix, as detailed in the Appendix D.

If we again take the derivatives with respect to the parameters of Equations (1) and (2) we obtain the Hessian. In total there are three expressions, one for each of the three blocks of the Hessian,

$$\frac{\partial^2 \mathcal{L}}{\partial W_{2k}\partial W_{2j}} = \frac{1}{N}\sum_{i=1}^{N} h_{ik}h_{ij} \ , \tag{3}$$

$$\frac{\partial^2 \mathcal{L}}{\partial W_{1pq}\partial W_{1mn}} = \frac{1}{N}\sum_{i=1}^{N} \left[ W_{2m}W_{2p}g'(z_{im})g'(z_{ip})x_{in}x_{iq} + (y_i - y_i^\star)\, W_{2m}g''(z_{im})x_{in}x_{iq}\delta_{pm}\right] \ , \tag{4}$$

$$\frac{\partial^2 \mathcal{L}}{\partial W_{2k}\partial W_{1mn}} = \frac{1}{N}\sum_{i=1}^{N} \left[ \delta_{km}\left(y_i - y_i^\star\right)g'(z_{im})x_{in} + h_{ik}W_{2m}g'(z_{im})x_{in}\right] \ . \tag{5}$$

To obtain analytical results, we need additional assumptions. We assume that the input data are distributed according to a standard normal distribution centered at the origin. With this in mind, we define the generalization error as the expected value of the loss function, given by

$$\mathcal{L}_g = \mathbb{E}_{\boldsymbol{x}}\left[\mathcal{L}\right] = \mathbb{E}_{\boldsymbol{x}}\left[\frac{1}{2}\left(y - y^\star\right)^2\right] = \mathbb{E}_{\boldsymbol{x}}\left[\frac{1}{2}\left(f_\theta(\boldsymbol{x}) - f_{\theta^*}(\boldsymbol{x})\right)^2\right] = \int_{\mathbb{R}^n}\frac{1}{2}\left(\frac{1}{2\pi}\right)^{d/2}\left(f_\theta(\boldsymbol{x}) - f_{\theta^*}(\boldsymbol{x})\right)^2 e^{-\frac{1}{2}\|\boldsymbol{x}\|^2}d\boldsymbol{x} \ .$$

The expressions for learning and Hessian components remain the same as long as we make the substitution $\frac{1}{N}\sum_{i=1}^{N} \to \int_{\mathbb{R}^n}\left(\frac{1}{2\pi}\right)^{d/2} e^{-\frac{1}{2}\|\boldsymbol{x}\|^2}d\boldsymbol{x}$. The advantage of working with the generalization error is that we are now able to perform analytical computations by taking expected values of Equations (3-5). These result in expressions that depend on the weights of the student network, which coincide with the weights of the teacher network at the optimal point. From this point onwards, we will only work with the generalization error, denoting it by $\mathcal{L}$ to simplify the notation. Similarly, unless stated otherwise, we use the designations loss function and generalization error interchangeably.

## 2.1 Learning Dynamics near the Optimal Point

Near the optimal point, $\theta^*$, which is the minimum of the loss function, $\mathcal{L}(\theta)$, with $\theta$ being the set of all trainable parameters of the network, the Hessian matrix determines the convergence rate of a network initialized at a point $\theta'$ close to $\theta^*$. To see this, we look at the gradient flow dynamics $\frac{d\theta}{dt} = -\frac{d\mathcal{L}}{d\theta}$ around $\theta^*$, by taking the Taylor series of $\mathcal{L}(\theta)$ around $\theta^*$,

$$\mathcal{L}(\theta) \approx \mathcal{L}(\theta^*) + \sum_{i=1}^{D}\frac{\partial \mathcal{L}(\theta^*)}{\partial \theta_i}\delta\theta_i + \frac{1}{2}\sum_{i,j=1}^{D}\frac{\partial^2\mathcal{L}(\theta^*)}{\partial\theta_i\partial\theta_j}\delta\theta_i\delta\theta_j$$

$$= \frac{1}{2}\sum_{i,j=1}^{D}\delta\theta_i H_{ij}\delta\theta_j \ ,$$

with $\delta\theta_i = (\theta_i - \theta_i^*)$, $H_{ij}$ being the $(i,j)$ Hessian matrix component, and where $D = \dim(\theta)$ is the number of parameters. The first two terms of the Taylor series vanish at the minimizer of the loss function, as it is a zero as well as an absolute minimum. Now, we have that

$$\frac{d\theta}{dt} = -\nabla_\theta \mathcal{L} \approx -H\delta\theta \implies \delta\theta(t) \approx \delta\theta(0)e^{-Ht} \ ,$$

so that the exponential of the Hessian controls how the network parameters evolve with time. If we know the eigensystem of $H$, i.e., the unit eigenvectors $v_i$ and respective eigenvalues $\lambda_i$ such that $Hv_i = \lambda_i v_i$, we have that

$$\|\delta\theta(t)\|^2 \approx \sum_{i=1}^{D} e^{-2\lambda_i t}\left|v_i^T\ \delta\theta(0)\right|^2 \ . \tag{6}$$

This shows that, ultimately, the eigenvalues near the optimum point of the loss function drive the parameter evolution under gradient descent.

We can lose the dependency on the eigenvectors in Equation (6) by averaging over the initial condition, $\delta\theta(0)$, assuming it follows a multivariate Gaussian distribution with mean zero and variance $\sigma_0^2$, yielding

$$\left\langle \|\delta\theta(t)\|^2 \right\rangle_{\delta\theta(0)} = \sum_{i=1}^{D} e^{-2\lambda_i t} \left\langle \left| v_i^T \cdot \delta\theta(0) \right|^2 \right\rangle_{\delta\theta(0)}$$

$$= \sigma_0^2 \sum_{i=1}^{T} e^{-2\lambda_i t} .$$

An expression for the time evolution of the loss function can thus be derived

$$\langle \mathcal{L}(t) \rangle_{\delta\theta(0)} \approx \frac{1}{2} \left\langle \delta\theta(0)^T e^{-2Ht} H \delta\theta(0) \right\rangle_{\delta\theta(0)}$$

$$= \frac{\sigma_0^2}{2} \operatorname{Tr} e^{-2Ht} H$$

$$= \frac{\sigma_0^2}{2} \int_0^{+\infty} e^{-2\lambda t} \lambda\rho(\lambda) d\lambda , \tag{7}$$

where $\rho(\lambda)$ is the eigenspectrum of the Hessian matrix.

For a single realization of the teacher-student setup, where the spectrum is discrete, we thus expect the large-time behavior of the loss function near the optimal point to be determined by the smallest non-zero eigenvalue. This behavior can be clearly seen in the linear network, discussed in the next section.

## 3 The Linear Network

In a linear network, the activation function is the identity function, i.e., we have $g(x) = x$, so that $g'(x) = 1$ and $g''(x) = 0$. Its first derivative is always equal to one and the second derivative is always equal to zero. Taking the second derivatives of the generalization error with respect to the student's weights leads to the following Hessian components at the optimal point,

$$\frac{\partial^2 \mathcal{L}}{\partial W_{2k} \partial W_{2j}} = E_x \left[ h_k h_j \right] = \sum_m W_{1km} W_{1jm} , \tag{8}$$

$$\frac{\partial^2 \mathcal{L}}{\partial W_{1pq} \partial W_{1mn}} = W_{2m} W_{2p} \delta_{nq} , \tag{9}$$

$$\frac{\partial^2 \mathcal{L}}{\partial W_{2k} \partial W_{1mn}} = W_{1kn} W_{2m} , \tag{10}$$

where we used the fact that the loss function is zero at the optimum. We find that, for linear networks, each sub-matrix of the Hessian depends only on weights of a single layer and only the anti-diagonal contains cross terms.

As these equations are valid at the optimal point, we have that the teacher's weights are the same as the student's weights. Thus, using Equations (9-10), together with the teacher weights, we can directly compute the Hessian matrix of a student at the optimal point. From the Hessian, we can then also calculate the eigenvalues. More interestingly, if we know how the weights of the teacher network are distributed, we can even derive an expression for the eigenvalue distribution of the Hessian.

### 3.1 Eigenvalues of the Hessian at the Optimal Point

The linear teacher-student setup exhibits a particularly interesting structure for the eigenvalues of the Hessian at the optimal point: they are given by sums of eigenvalues of each of the diagonal sub-matrices of the Hessian. For a detailed analysis we refer the reader to Appendix A.

Thus, in the linear network, we are able to calculate the entire eigenspectrum from the eigenspectrum of each diagonal sub-matrix of the Hessian. The **A** sub-matrix has just one non-zero eigenvalue given by

$\sum_{k=1}^{N_h} W_{2k}^2$. This eigenvalue has a multiplicity equal to $N_i$, the input dimension. Under the assumption that the components of $\boldsymbol{W}_2$ were taken from a normal distribution with zero mean and variance $\frac{1}{N_h}$, we find that this eigenvalue follows a scaled chi-squared distribution with $N_h$ degrees of freedom, $\lambda \sim \frac{1}{N_h}\chi_{N_h}^2$. On the other hand, the $\boldsymbol{B}$ block has $\min(N_i, N_h)$ non-zero eigenvalues that asymptotically follow a scaled Marchenko-Pastur distribution (Götze & Tikhomirov, 2004). The probability density function of the scaled Marchenko-Pastur distribution is given by Equation (15) in Appendix A.

The eigenvalues of the entire Hessian matrix are given by summing the eigenvalue of the upper left block with the (possibly zero-padded) eigenvalues of the lower right block, yielding $N_i$ non-zero eigenvalues in total. Thus, if $N_i \leq N_h$, asymptotically the eigenvalues follow a convolution of the scaled chi-square distribution with the scaled Marchenko-Pastur distribution, denoted by $\mathcal{C}$. If $N_i > N_h$, then the eigenvalue distribution is a mixed distribution, asymptotically given by

$$\lambda \sim \frac{N_i - N_h}{N_i}\left(\frac{1}{N_h}\chi_{N_h}^2\right) + \frac{N_h}{N_i}\mathcal{C} \ .$$

Figure 2 illustrates both cases. On the left, we have $N_i < N_h$, thus the eigenvalue distribution is fully described by the convolution of the scaled chi-squared distribution and the scaled Marchenko-Pastur distribution. On the right, we have $N_i > N_h$, thus the distribution is mixed, having contributions from both the convoluted distribution as well as the scaled chi-squared distribution.

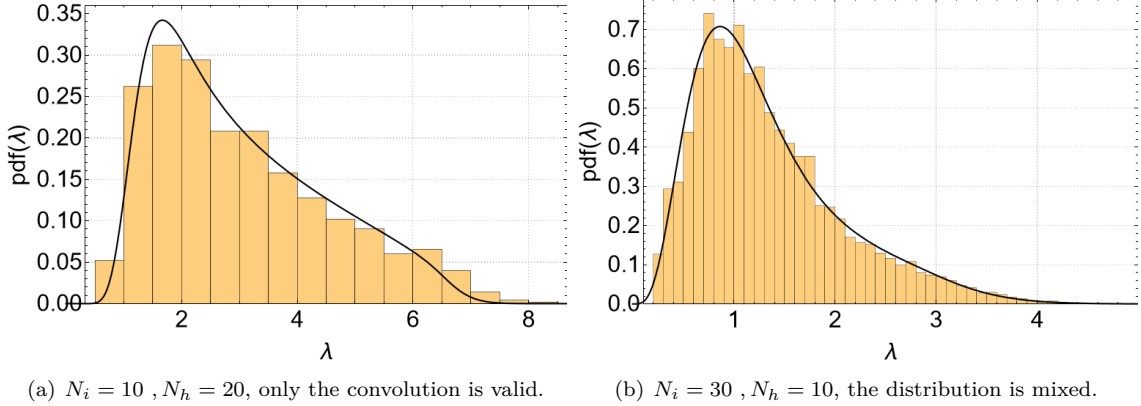

(a) $N_i = 10$ , $N_h = 20$, only the convolution is valid.    (b) $N_i = 30$ , $N_h = 10$, the distribution is mixed.

Figure 2: Agreement between the predicted eigenvalue distribution for the Hessian and numerical simulations.

A detailed proof for the expression of the Hessian eigenspectrum in the linear teacher-student setup can be found in Appendix A. This eigenvalue distribution is quite different from all the others distributions considered in the present work. The most notable difference is that, unlike non-linear networks, here there is a lack of a bulk of eigenvalues concentrated near zero. As such, the long term behavior of Equation (7) can be more easily checked in this case, as we can clearly distinguish the correct eigenvalues from numerical error when we diagonalize the Hessian matrix that is determined by a test dataset.

Figure 3 shows, in blue, the loss function evolution with time of a linear student network initialized near the optimal point. Time here is to be understood as the product of the number of iterations with the learning rate. We obtain this by randomly generating a teacher network, copying its weights to the student, adding random noise to them, and finally training the student using stochastic gradient descent. In black, we see the function $\alpha \exp\left(-2\lambda_{min}t\right)$, where $\lambda_{min}$ is the smallest non-zero eigenvalue at the optimal point, calculated beforehand. Here, $\alpha$ is such that $\mathcal{L}(t_f) = \alpha \exp\left(-2\lambda_{min}t_f\right)$ so that both functions meet at the final time, $t_f$. Initially the loss function decays faster due to the contribution from the larger eigenvalues. However, the long-time behavior of the loss function is well determined by just the exponential of its smallest eigenvalue, as predicted.

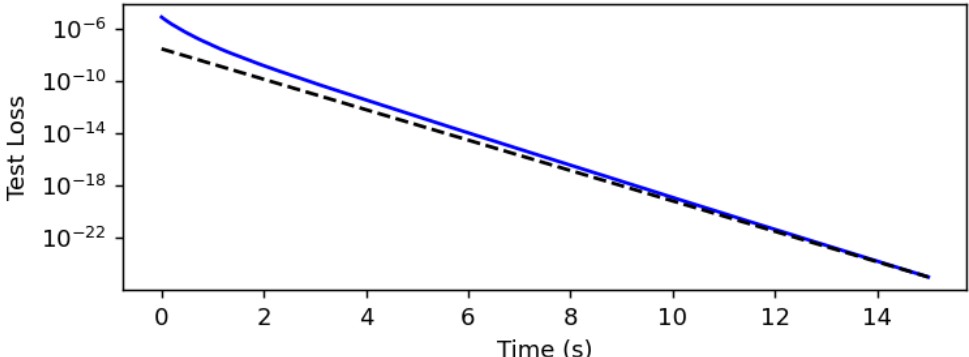

Figure 3: Agreement between the loss function of a student network initialized near the optimum point and the exponential of the smallest eigenvalue.

### 3.2 Number of Effective Parameters

In the linear network, we find that for a randomly generated teacher with weights following a normal distribution, the Hessian at the optimal point always has $N_i$ positive eigenvalues, with all others being zero. Thus, in terms of the loss landscape, of the $(N_i + 1)N_h$ possible directions in which the loss can vary, only a set of $N_i$ directions are required to fully characterize the behavior of the loss function around the optimal point. The loss function does not vary in all the other directions orthogonal to these $N_i$ directions.

We can look at this from another perspective, trying to answer the following question: "When is a student network equivalent to another student network?". Here, "equivalent" is to be understood as, given any input, both students giving the exact same output. To answer this question, we notice that, due to the activation function being the identity, the student network takes the form

$$y = \boldsymbol{W}_2 \boldsymbol{W}_1 \boldsymbol{x} = \boldsymbol{A} \boldsymbol{x} \ ,$$

where $\boldsymbol{A}$ is a $1 \times N_i$ matrix. Therefore, for two students to be equivalent, it is only required that the components of $\boldsymbol{A}$ are equal. This shows that, even though the network has $(N_i + 1)N_h$ parameters, due to the architecture, it can be encoded using only $N_i$ parameters.

Thus, for the linear network, we find that the number of positive eigenvalues of the Hessian, or equivalently, the rank of the Hessian, at the optimal point, gives a measure of an effective number of parameters.

## 4 Polynomial Networks

### 4.1 The Quadratic Network

In this subsection, we consider a teacher-student setup, where the underlying activation function is of the form

$$g(x) = x + \epsilon x^2 \ .$$

The associated first and second derivatives are $g'(x) = 1 + 2\epsilon x$ and $g''(x) = 2\epsilon$.

Following the same procedure as for the linear network, we solve the associated Gaussian integrals and arrive at the following expressions for the Hessian matrix components:

$$\frac{\partial^2 \mathcal{L}}{\partial W_{2k} \partial W_{2j}} = \sum_m W_{1km} W_{1jm} + \epsilon^2 F_1(k, j) \;, \tag{11}$$

$$\frac{\partial^2 \mathcal{L}}{\partial W_{1pq} \partial W_{1mn}} = W_{2m} W_{2p} \delta_{nq} + 4\epsilon^2 F_2(p, q, m, n) \;, \tag{12}$$

$$\frac{\partial^2 \mathcal{L}}{\partial W_{2k} \partial W_{1mn}} = W_{2m} W_{1kn} + 2\epsilon^2 F_3(k, m, n) \;, \tag{13}$$

with

$$F_1(k, j) = \sum_m W_{1km}^2 \sum_m W_{1jm}^2 + 2 \left( \sum_m W_{1km} W_{1jm} \right)^2 \;,$$

$$F_2(p, q, m, n) = \delta_{nq} \sum_r W_{1mr} W_{1pr} + W_{1mn} W_{1pq} + W_{1mq} W_{1pn} \;,$$

$$F_3(k, m, n) = W_{1mn} \sum_p (W_{1kp})^2 + 2W_{1kn} \sum_p W_{1mp} W_{1kp} \;.$$

As expected, for $\epsilon = 0$, we recover Equations (9-10) for the linear network. However, unlike the linear case, here we were not able to analytically obtain the eigenvalues. Nevertheless, we briefly make some remarks on the eigenspectrum of quadratic networks.

Numerically, the eigenspectrum of a realization of this teacher-student setup where the weights of the teacher are sampled from a normal distribution exhibits a bulk of eigenvalues near zero, with a single eigenvalue being much farther apart from the bulk. In Figure 4, we see that behavior when the input dimension equals the hidden dimension: as the dimension of the network increases, the bulk of the eigenvalues gets closer to zero. We note that the gaps between lines of the same color represent areas where we did not observe any eigenvalues empirically. In opposition, the highest eigenvalue, which is always far away from the bulk near zero, diverges, leaving another much smaller bulk of eigenvalues for larger and larger values of $\lambda$, as the dimensions of the network increase.

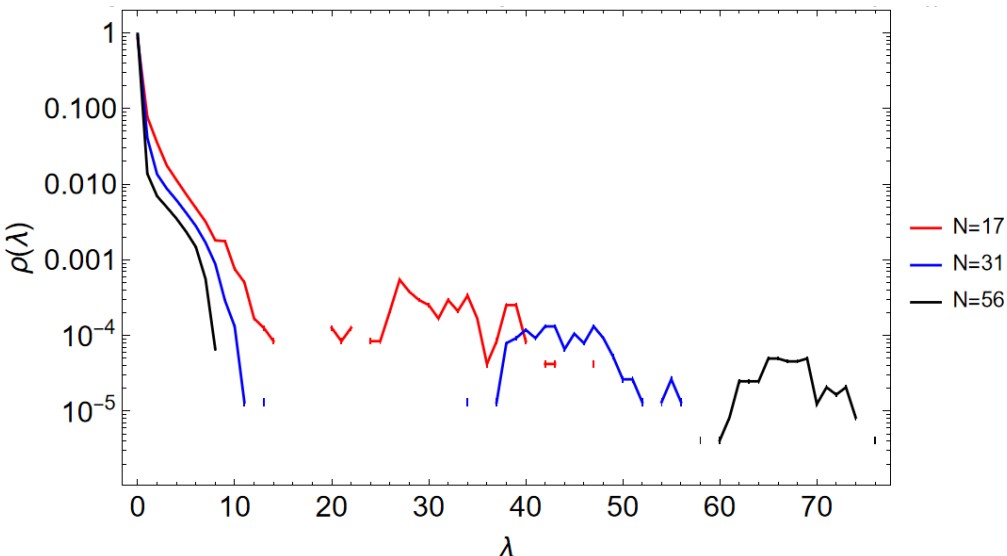

Figure 4: Eigenspectrum distribution for a quadratic teacher-student setup with $\epsilon = 1$ and $N_i = N_h = N$.

### 4.1.1 Number of Effective Parameters

In the linear case, we saw that a network with $N_i$ input neurons and $N_h$ hidden neurons has $(N_i + 1)N_h$ parameters in total, but its Hessian matrix has at most $N_i$ non-zero eigenvalues. In the quadratic case, if $N_h$ is the number of hidden neurons, then above a certain threshold, say $T$, for $N_h > T$ the number of zeroes of the Hessian is given by $(N_h - 1) N_h/2$, which are the triangular numbers $(0, 1, 3, 6, 10, 15, 21, 28, \ldots)$.

We use the same strategy as in the linear network to find the number of effective parameters, $N_{\text{eff}}$. As the activation function is only used once, in the hidden layer, we know that the function being represented by the student network is a quadratic function of the input vector $\boldsymbol{x}$. Such a function can be written as

$$f(\vec{x}) = \boldsymbol{A} \cdot \boldsymbol{x} + \boldsymbol{x}^T \boldsymbol{B} \boldsymbol{x} \ ,$$

without a constant term as the network has no biases. Such a quadratic function is represented by $N_i$ parameters for $\boldsymbol{A}$ and $N_i(N_i + 1)/2$ for $\boldsymbol{B}$. The latter come from the fact that, in the expression $\boldsymbol{x}^T \boldsymbol{B} \boldsymbol{x} = \sum_{i,j=0}^{N} x_i x_j B_{ij}$, for $i \neq j$, what matters is the sum $B_{ji} + B_{ij}$, and so we only need to deal with symmetric matrices.

As such, the number of effective parameters of the quadratic network is $N_i + \frac{N_i(N_i+1)}{2}$. Note however, that this assumes that the network has enough free parameters in the Hessian matrix, in particular it is only valid whenever $N_h \geq N_i$.

The previous result is only an upper bound for the total number of independent parameters. The actual number depends on both $N_i$ and $N_h$, because the network may not be able to fully express any degree-two polynomial in the weights if $N_h$ is not large enough. Numerically, we found that the threshold for this upper bound to be saturated happens for $N_h = N_i$. We were able to prove that the number of effective parameters in the general case is given by

$$N_{\text{eff}} = N_i + \frac{N_i(N_i + 1)}{2} - \left(\frac{\nu^2 + 3\nu}{2}\right) \text{H}(\nu) \ , \tag{14}$$

where $\nu = N_i - N_h$ and $\text{H}(x)$ is the Heaviside step function. We refer to Appendix B for a detailed proof of Equation (14).

## 4.2 Upper Bound for the Effective Number of Parameters for Higher-Order Polynomials

We can generalize the analysis above for activation functions that are higher-order polynomials. We can give an upper bound for the effective number of parameters by counting the number of parameters that describe an $n$-th polynomial function of a vector $\boldsymbol{x}$. Similarly to the quadratic network approach, a degree-$n$ polynomial can be represented as

$$f(\vec{x}) = \sum_{i=1}^{N_i} A_i^{(1)} x_i + \sum_{i,j=1}^{N_i} A_{ij}^{(2)} x_i x_j + \cdots + \sum_{i_1,\ldots,i_n=1}^{N_i} A_{i_1 \ldots i_n}^{(n)} x_{i_1} \cdots x_{i_n} \ ,$$

where each $A^{(k)}$ is a $k$-dimensional symmetric tensor. Finally, counting the number of parameters amounts to summing the number of symmetric components of each tensor. For a symmetric tensor of order $n$, the number of independent parameters is given by $\binom{N_i+n-1}{n}$. With the help of the hockey-stick identity (Jones, 1996), we find that

$$N_{\text{eff}} \leq \sum_{k=1}^{n} \binom{N_i + k - 1}{k} = \binom{N_i + n}{n} - 1 \ ,$$

which is the upper bound for the number of effective parameters for a polynomial activation function of degree $n$.

## 5  Error Function Network

Finally. we study the case where the activation function is the error function

$$g(x) = \text{erf}\left(\frac{x}{\sqrt{2}}\right) = \frac{2}{\sqrt{\pi}} \int_0^{x/\sqrt{2}} e^{-t^2} dt \ ,$$

which has first derivative $g'(x) = \sqrt{2/\pi} \ e^{-x^2/2}$ and second derivative $g''(x) = -x \ g'(x)$. This choice of activation function is common in machine learning as it is a sigmoid function. Moreover, its associated Gaussian integrals are well defined analytically. The derivation and final system of Equations (16-18) is available in Appendix C.

The results of a numerical analysis of the eigenspectrum of the Hessian matrix obtained from Equations (16-18) are displayed in Figure 5. We see a similar structure to the quadratic case. There is a bulk of positive eigenvalues near zero that, as the network size increases, gets closer and closer to zero. There is another bulk of larger eigenvalues that remains fixed as the network size increases. For a single realization of this teacher-student setup, we find that the number of eigenvalues belonging to the bulk away from zero is always equal to $N_h$, the number of parameters in $\boldsymbol{W}_2$. This may be due to the fact that the output layer does not have an activation function whereas the hidden layer does.

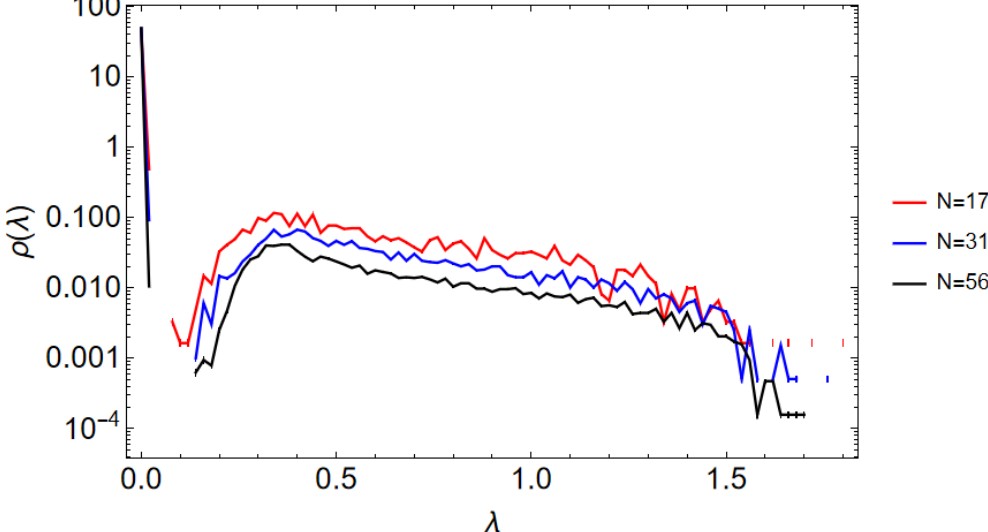

Figure 5: Eigenspectrum distribution for a error function teacher-student setup with $N_i = N_h = N$.

### 5.1  Number of Effective Parameters

For the error function network, we empirically find that no matter how large the networks are, the Hessian matrix is always full rank. If we try to follow our strategy for counting the number of effective parameters, as the teacher network is some linear combination of error functions, we can no longer compress this network onto a finite polynomial. Thus, it appears that no further *compression* is possible and the entire suite of parameters of the teacher is required for the student to perfectly replicate the teacher's outputs.

## 6  Conclusion and Future Work

In this work, we explored the Hessian rank as a measure of the effective number of parameters. Under the teacher-student setup, where the teacher and the student networks share their architecture, we found that this measure is valid for polynomial networks, for a sufficiently large number of hidden neurons. For a more complicated activation function, like the error function, this approach breaks down as the Hessian is always

full rank at the optimal point, an indication that no further compression of the number of parameters can be done for such networks.

For all linear, quadratic, and error function networks, we were able to derive the equations for the Hessian components, under the assumption of the input distribution being a Gaussian centered at zero. Specifically for linear networks, we were able to analytically derive the distribution of the eigenvalues of the Hessian, when the distribution of weights of the teacher network is known. In the case where this distribution is Gaussian, we found that, in general, the distribution of eigenvalues of the Hessian asymptotically follows a convolution of a scaled Marchenko-Pastur distribution with a scaled chi-squared distribution.

Finally, we point out future research directions in this teacher-student perspective. Firstly, we could study how this notion of an effective number of parameters changes if we add biases to the analysis. Following the same strategy as in this work, we would expect to obtain the same results incremented by one. This new degree of freedom comes from the constant term of the polynomial function in the compressed representation. Following the same train of thought, we could also try to generalize the result for deeper networks. On the other hand, still working with two-layer neural networks, one can possibly study the effects of over-parameterization on student networks. This is because, in the case where the student has more parameters than the teacher, the optimal solution still exists. The optimal solution in this case can be obtained by zeroing out any extra neurons present in the student network. Comparing the Hessian spectrum of such an overparameterized student with the ones obtained in this work could provide insight into the unreasonable performance of large neural networks. Finally, one other possible path to explore would be to study the Hessian rank evolution with time.

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

# A  The Linear Teacher-Student Case

To understand where the expression for the distribution of the eigenvalues of the Hessian comes from, it is helpful to first study the eigenvalues of each Hessian sub-matrix separately.

## A.1  Eigenvalues of the $A$ block of the Hessian

The $A$ block of the Hessian matrix has an interesting symmetry to it. Its components are given by Equation (9),

$$\frac{\partial^2 \mathcal{L}}{\partial W_{1pq} \partial W_{1mn}} = W_{2m} W_{2p} \delta_{nq} \; .$$

This matrix is very sparse. We flatten the matrix $W_1$ by following a row-first convention, i.e., we flatten the matrix by concatenating its rows. As such, the $\delta_{nq}$ term makes it so that we can only have a non-zero element every $N_h$ steps, when $k \bmod N_h = 0$, with $k$ being the current row. The effect of this term is that this matrix will be built of identity $I_{N_i}$ blocks being multiplied by a real number.

For example, for $N_i = 3$ and $N_h = 4$ we would have

$$\begin{pmatrix}
\mathrm{W2}_1^2 & 0 & 0 & \mathrm{W2}_1\mathrm{W2}_2 & 0 & 0 & \mathrm{W2}_1\mathrm{W2}_3 & 0 & 0 & \mathrm{W2}_1\mathrm{W2}_4 & 0 & 0 \\
0 & \mathrm{W2}_1^2 & 0 & 0 & \mathrm{W2}_1\mathrm{W2}_2 & 0 & 0 & \mathrm{W2}_1\mathrm{W2}_3 & 0 & 0 & \mathrm{W2}_1\mathrm{W2}_4 & 0 \\
0 & 0 & \mathrm{W2}_1^2 & 0 & 0 & \mathrm{W2}_1\mathrm{W2}_2 & 0 & 0 & \mathrm{W2}_1\mathrm{W2}_3 & 0 & 0 & \mathrm{W2}_1\mathrm{W2}_4 \\
\mathrm{W2}_1\mathrm{W2}_2 & 0 & 0 & \mathrm{W2}_2^2 & 0 & 0 & \mathrm{W2}_2\mathrm{W2}_3 & 0 & 0 & \mathrm{W2}_2\mathrm{W2}_4 & 0 & 0 \\
0 & \mathrm{W2}_1\mathrm{W2}_2 & 0 & 0 & \mathrm{W2}_2^2 & 0 & 0 & \mathrm{W2}_2\mathrm{W2}_3 & 0 & 0 & \mathrm{W2}_2\mathrm{W2}_4 & 0 \\
0 & 0 & \mathrm{W2}_1\mathrm{W2}_2 & 0 & 0 & \mathrm{W2}_2^2 & 0 & 0 & \mathrm{W2}_2\mathrm{W2}_3 & 0 & 0 & \mathrm{W2}_2\mathrm{W2}_4 \\
\mathrm{W2}_1\mathrm{W2}_3 & 0 & 0 & \mathrm{W2}_2\mathrm{W2}_3 & 0 & 0 & \mathrm{W2}_3^2 & 0 & 0 & \mathrm{W2}_3\mathrm{W2}_4 & 0 & 0 \\
0 & \mathrm{W2}_1\mathrm{W2}_3 & 0 & 0 & \mathrm{W2}_2\mathrm{W2}_3 & 0 & 0 & \mathrm{W2}_3^2 & 0 & 0 & \mathrm{W2}_3\mathrm{W2}_4 & 0 \\
0 & 0 & \mathrm{W2}_1\mathrm{W2}_3 & 0 & 0 & \mathrm{W2}_2\mathrm{W2}_3 & 0 & 0 & \mathrm{W2}_3^2 & 0 & 0 & \mathrm{W2}_3\mathrm{W2}_4 \\
\mathrm{W2}_1\mathrm{W2}_4 & 0 & 0 & \mathrm{W2}_2\mathrm{W2}_4 & 0 & 0 & \mathrm{W2}_3\mathrm{W2}_4 & 0 & 0 & \mathrm{W2}_4^2 & 0 & 0 \\
0 & \mathrm{W2}_1\mathrm{W2}_4 & 0 & 0 & \mathrm{W2}_2\mathrm{W2}_4 & 0 & 0 & \mathrm{W2}_3\mathrm{W2}_4 & 0 & 0 & \mathrm{W2}_4^2 & 0 \\
0 & 0 & \mathrm{W2}_1\mathrm{W2}_4 & 0 & 0 & \mathrm{W2}_2\mathrm{W2}_4 & 0 & 0 & \mathrm{W2}_3\mathrm{W2}_4 & 0 & 0 & \mathrm{W2}_4^2
\end{pmatrix}$$

We can rewrite the above matrix in a more compact notation using the following tensor product

$$\begin{pmatrix}
\mathrm{W2}_1^2 & \mathrm{W2}_1\mathrm{W2}_2 & \mathrm{W2}_1\mathrm{W2}_3 & \mathrm{W2}_1\mathrm{W2}_4 \\
\mathrm{W2}_1\mathrm{W2}_2 & \mathrm{W2}_2^2 & \mathrm{W2}_2\mathrm{W2}_3 & \mathrm{W2}_2\mathrm{W2}_4 \\
\mathrm{W2}_1\mathrm{W2}_3 & \mathrm{W2}_2\mathrm{W2}_3 & \mathrm{W2}_3^2 & \mathrm{W2}_3\mathrm{W2}_4 \\
\mathrm{W2}_1\mathrm{W2}_4 & \mathrm{W2}_2\mathrm{W2}_4 & \mathrm{W2}_3\mathrm{W2}_4 & \mathrm{W2}_4^2
\end{pmatrix} \otimes I_3 \; .$$

One important consequence of the above is that every eigenvalue of the original matrix will have a multiplicity of $N_i$. In general, this block of the Hessian matrix can be written as $(\boldsymbol{W_2}\boldsymbol{W_2^T}) \otimes I_{N_i}$, with $\boldsymbol{W_2} \in \mathbb{R}^{N_h}$. Thus, $\boldsymbol{W_2}$ has $N_h - 1$ orthogonal directions with eigenvalue equal to zero. To get the other eigenvalue, note that by associativity

$$(\boldsymbol{W_2}\boldsymbol{W_2^T})\boldsymbol{W_2} = ||\boldsymbol{W_2}||^2 \boldsymbol{W_2} \; .$$

From this we get that the other eigenvalue is given by $\sum_{i=1}^{N_h} W_{2i}^2$. We thus have the values and multiplicities of all eigenvalues of the $A$ block of the Hessian matrix: we have $N_i(N_h - 1)$ eigenvalues equal to zero and $N_i$ equal to $\sum_{i=1}^{N_h} W_{2i}^2$. Being sums of squares of a normal distribution with fixed variance, this eigenvalue follows a scaled chi-square distribution with $N_h$ degrees of freedom, $\lambda \sim \sigma^2 \chi_{N_h}^2$. In the following figure we can see some numerical simulations of this part of the Hessian as well as the predicted distribution on top.

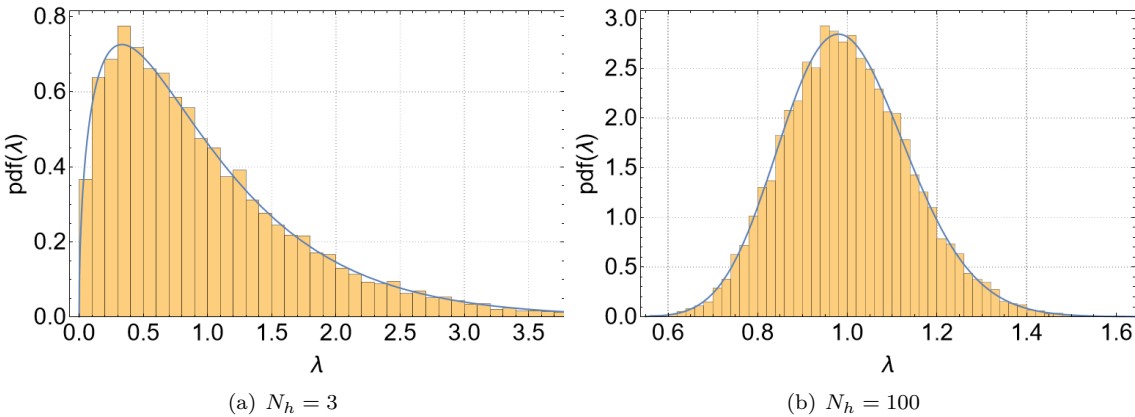

(a) $N_h = 3$         (b) $N_h = 100$

Figure 6: Agreement between the predicted distribution for this eigenvalue of the first diagonal block, $\boldsymbol{A}$, of the Hessian and numerical simulations.

### A.2   Eigenvalues of the $B$ block of the Hessian

This block is given by the expression $\boldsymbol{W_1}\boldsymbol{W_1^T}$. Asymptotically, the eigenvalues of this block then follow a scaled Marchenko-Pastur distribution (because the entries are not being normalized) (Götze & Tikhomirov, 2004). This distribution is scaled by a factor of $N_i$. Furthermore, if $N_h > N_i$ the matrix is singular and has only $N_i$ non-zero eigenvalues. In the other cases it has $N_h$ non-zero eigenvalues.

The pdf of the scaled distribution of eigenvalues is given by

$$pdf(x) = \frac{1}{2\pi\sigma^2}\frac{\sqrt{(\lambda_+ - x)(x - \lambda_-)}}{\lambda x}\mathbf{1}_{x\in[\lambda_-,\lambda_+]}\left(\mathbf{1}_{\lambda\leq 1} + \lambda\mathbf{1}_{\lambda>1}\right) \;, \tag{15}$$

where $\lambda = N_h/N_i$, $\lambda_\pm = \sigma^2\left(1\pm\sqrt{\lambda}\right)^2$, $\sigma^2 = \frac{1}{N_i}$ and $\mathbf{1}_C$ is an indicator function valid in the region denoted by the condition $C$. In the following figure we can see the agreement between the above predicted distribution and numerical experiments.

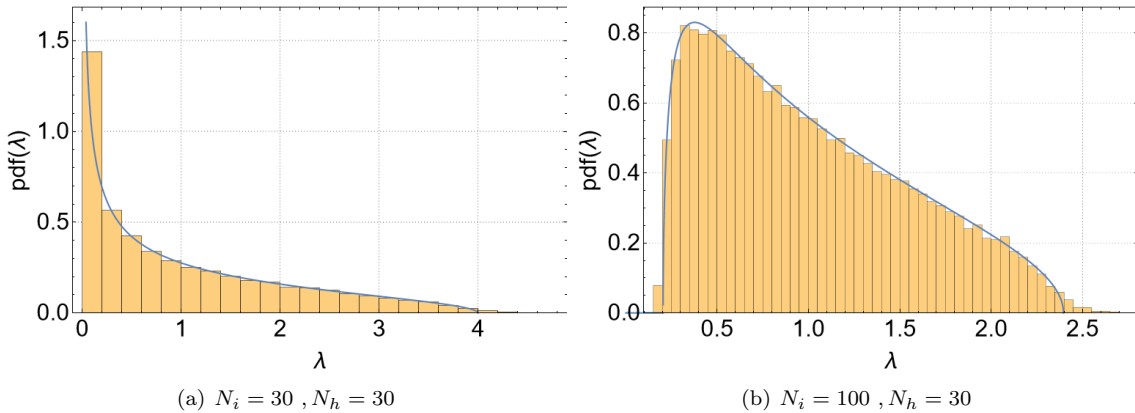

(a) $N_i = 30 \;, N_h = 30$        (b) $N_i = 100 \;, N_h = 30$

Figure 7: Agreement between the predicted eigenvalue distribution for the second diagonal block, $\boldsymbol{B}$, of the Hessian and numerical simulations.

### A.3   A description of the positive eigenvalues for the Hessian

The Hessian matrix may be written as

$$H = \begin{pmatrix} \boldsymbol{W_2}\boldsymbol{W_2^T} \otimes I_{N_i} & \boldsymbol{W_2} \otimes \boldsymbol{W_1^T} \\ \boldsymbol{W_2^T} \otimes \boldsymbol{W_1} & \boldsymbol{W_1}\boldsymbol{W_1^T} \end{pmatrix} \ .$$

We have that $\boldsymbol{W_2}$ is a column vector and, as analysed previously, its eigenvector is $v := \boldsymbol{W_2^T}$, we shall call the associated eigenvector $\lambda_2$. From the rank-nullity theorem for $W_1$, we have that $N_i = \text{rank}\,(W_1) + \text{null}\,(W_1)$. From the SVD decomposition of $W_1$, we have that there will be as many non-zero singular values as the rank of $W_1$. For each one of these non-zero singular values, call it $\sqrt{\lambda_1}$, let $z'$ be the corresponding singular vector. Then we have that

$$\begin{cases} W_1 z' = \sqrt{\lambda_1} z \\ W_1^T z = \sqrt{\lambda_1} z' \end{cases} ,$$

which implies that $z$ is an eigenvector of $W_1 W_1^T$ associated with the eigenvalue $\lambda_1$.

Now, note that, with the vector given by $\boldsymbol{y} = [(v \otimes z')\ \ \sqrt{\lambda_1}z]^T$, the product of the Hessian matrix with the above vector gives $\boldsymbol{Hy} = (\lambda_1 + \lambda_2)\boldsymbol{y}$. Thus, for each non-zero singular value of $W_1$ we have obtained an eigenvector of the Hessian where the corresponding eigenvalue is indeed the sum of the eigenvalues of each diagonal block matrix of the Hessian. The number of such eigenvectors is equal to the rank of $W_1$.

Now, let $z'$ be a vector in the kernel of $W_1$. The above relations remain true if we take into account that now $\lambda_1 = 0$. And thus the vector $\boldsymbol{y} = \begin{bmatrix} (v \otimes z') & \vec{0} \end{bmatrix}^T$, where $\vec{0}$ is a vector of $N_h$ zeros, is also an eigenvector of $\boldsymbol{H}$, as $\boldsymbol{Hy} = \lambda_2 \boldsymbol{y} = (0 + \lambda_2)\,\boldsymbol{y} = (\lambda_1 + \lambda_2)\boldsymbol{y}$. For the random matrices we considered for the teacher, the kernel of $W_1$ is non-trivial almost surely whenever $N_i \geq N_h$.

Thus, in general, by the rank-nullity theorem for $W_1$, we will have $N_i$ linearly independent eigenvectors of the Hessian matrix where the associated eigenvalues are the sum of the eigenvalues of each diagonal submatrix of the Hessian.

# B Counting the Effective Parameters of a Polynomial Network

## B.1 Quadratic Network: The $N_h < N_i$ case

To make this discussion more clear, we shall consider the case where $N_h = N_i - 1$. Here, two interesting effects happen: the first is that both the vector $\vec{A}$ and the matrix $\boldsymbol{B}$ lose a dependency on the last parameter $W_2$, which accounts for the loss of one of the degrees of freedom, when compared to the upper bound. If we keep reducing $N_h$ the number of degrees of freedom should also decrease linearly, for the same reason.

The second effect is on the rank of the matrix $\boldsymbol{B}$. In the case where $N_i \leq N_h$, the matrix $\boldsymbol{B}$ is made up of the sum of at least $N_i$ random rank one matrices and so, in general, it will be a rank $N_i$ symmetric matrix. However, when we have $N_h < N_i$, now $\boldsymbol{B}$ is made up of only $N_h$ rank one matrices and, as such, it has a smaller number of independent parameters.

To count this new number, we note that any symmetric real matrix has an eigenvalue decomposition such that

$$\boldsymbol{B} = \boldsymbol{X}\boldsymbol{\Lambda}\boldsymbol{X}^{T} \ ,$$

where $\boldsymbol{\Lambda}$ is a diagonal matrix whose components are the eigenvalues of $\boldsymbol{B}$ and $\boldsymbol{X}$ is an $N_i \times N_i$ real orthogonal matrix. Furthermore, for real symmetric matrices, the number of non-zero eigenvalues is equal to the rank of the matrix.

If we assume that $\boldsymbol{B}$ has full rank, then the degrees of freedom of $\boldsymbol{B}$ can be derived from the following exercise:

- Each non-zero eigenvalue in $\boldsymbol{\Lambda}$ is one additional degree of freedom, giving $N$ in total.

- The orthogonal matrix $\boldsymbol{X}$ has $\frac{N_i(N_i-1)}{2}$ independent components.

- In total we thus have $N_i + \frac{N_i(N_i-1)}{2} = \frac{N_i(N_i+1)}{2}$.

Thus, we get the same number that we obtained before for the independent components of $\boldsymbol{B}$, if it is a full rank matrix.

Now, if $\boldsymbol{B}$ is a matrix of rank $r < N_i$, the following happens:

- We now have only $r$ degrees of freedom from the eigenvalues, as the other $N_i - r$ eigenvalues are zero.

- The orthogonal matrix has additional symmetries that appear due to the degeneracy of the zero eigenvalues. In particular, we need to remove the number of permutations of columns of $\boldsymbol{X}$ associated with the null eigenvalue, as they will yield the same matrix $\boldsymbol{B}$. Thus, we must remove $\frac{(N_i-r)(N_i-r-1)}{2} = \binom{N_i-r}{2}$ degrees of freedom.

- Another way to state the above is that the space of the zero eigenvectors of $\boldsymbol{B}$ does not contribute towards the number of independent components and, as such, we need to remove the number of degrees of freedom that those eigenvectors used to contribute. That happens to be $\binom{N_i-r}{2}$.

- Yet another way to put is to say that the matrix $\boldsymbol{B}$ is independent of rotations of the space associated with the null eigenvectors as, as such, we need to remove $\frac{n(n-1)}{2}$ degrees of freedom from the orthogonal matrix.

- Thus, when $\boldsymbol{B}$ is a rank $r$ matrix we have

$$r + \frac{N_i\left(N_i - 1\right)}{2} - \binom{N_i - r}{2}$$

degrees of freedom.

The overall result is that, whenever $r = N_h < N_i$, the number of effective parameters of the network is given by

$$N_i + \frac{N_i(N_i+1)}{2} - (N_i - r) - (N_i - r + \binom{N_i - r}{2})) = N_i + \frac{N_i(N_i+1)}{2} - \left(\frac{(N_i-r)^2 + 3(N_i-r)}{2}\right) \,,$$

which yields the predicted numerical results.

### B.2 The number of symmetric components of a tensor of order $n$

Let us recall here how to obtain the number of symmetric components of a tensor of order $n$. Let $A^{(n)}$ be a symmetric tensor of order $n$ where the size of each dimension is $N_i$. Then $\binom{N_i}{n}$ gives the number of distinct groupings of $n$ indices, thus accounting for the fact that the tensor is symmetric.

However, we are missing the terms with repeated indices. One way to account for them is to introduce some abstract symbol $R_1$ to indicate the repetition of some element.

For example, in counting the number of independent components of a symmetric matrix, $\binom{N_i+1}{2}$ represents all the possible pairings of the $N_i$ different values for each index as well as the additional $R_1$ symbol, which we must interpret as the instruction "repeat the 1st lowest number in this pairing". Thus, $\{1, R_1\} \leftrightarrow x_1 x_1$ whereas $\{1, 2\} \leftrightarrow x_1 x_2 \leftrightarrow x_2 x_1$.

To generalize this to higher dimensions we just need to add additional repetition symbols that are interpreted as "$R_n$ means that the $n$ th lowest number in this grouping is to be repeated once". Thus, in the cubic case we interpret $\{1, R_1, 2\} \leftrightarrow x_1 x_1 x_2$ and $\{R_2, 1, 2\} \leftrightarrow x_1 x_2 x_2$ – giving us the components of the tensor with two equal indices. Also, we interpret $\{R_1, 2, R_2\} \leftrightarrow x_2 x_2 x_2$, which gives the components with three repeated indices. With the addition of these symbols we can see that the expression $\binom{N_i+n-1}{n}$ counts all independent symmetric components of an $n$-th order tensor.

## C  The Error Function Network Hessian equations

The equations for the case where the activation function is the error function are:

$$\frac{\partial \mathcal{L}}{\partial W_{1pq} \partial W_{1mn}} = \frac{2 W_{2m} W_{2p}}{\pi \sqrt{\det \Sigma^{-1}}} [\Sigma]_{nq} \ , \tag{16}$$

$$\frac{\partial^2 \mathcal{L}}{\partial W_{2k} \partial W_{2j}} = \frac{2}{\pi} \arctan \left( \frac{\sum_{i=1}^{n} W_{ji} W_{ki}}{\sqrt{1 + \sum_{i=1}^{n} W_{ki}^2 + \sum_{i=1}^{n} W_{ji}^2 + \left( \sum_{i=1}^{n} W_{ki}^2 \right) \left( \sum_{i=1}^{n} W_{ji}^2 \right) - \left( \sum_{i=1}^{n} W_{ji} W_{ki} \right)^2}} \right) \ , \tag{17}$$

$$\frac{\partial^2 \mathcal{L}}{\partial W_{2k} \partial W_{1mn}} = \frac{2}{\pi \sqrt{1 + W_{1m} W_{1m}^T}} W_{2m} \frac{W_{1kn} - \frac{W_{1k} W_{1m}^T W_{1mn}}{1 + W_{1m} W_{1m}^T}}{\sqrt{1 + W_{1k} W_{1k}^T - \frac{(W_{1m} W_{1k}^T)^2}{1 + W_{1m} W_{1m}^T}}} \ , \tag{18}$$

where we have that

$$\Sigma = I - \frac{W_m^T W_m}{1 + W_m W_m^T} - \frac{W_p^T W_p}{1 + W_p W_p^T} - \left( \frac{(W_m W_p^T)^2 \left( \frac{W_m^T W_m}{1 + W_m W_m^T} + \frac{W_p^T W_p}{1 + W_p W_p^T} \right) - (W_m W_p^T)(W_m^T W_p + W_p^T W_m)}{(1 + W_p W_p^T)(1 + W_m W_m^T) - (W_m W_p^T)^2} \right)$$

.

**The Upper Left block**  Looking at the upper left part of the Hessian, which contains the $W_1 - W_1$ correlations, we need to calculate the following expected value

$$\frac{\partial \mathcal{L}}{\partial W_{1pq} \partial W_{1mn}} = W_{2m} W_{2p} E_x[g'(z_m) g'(z_p) x_n x_q] + W_{2m} \delta_{pm} E_x[g''(z_m) x_n x_q]$$

Focusing on the first additive term we have that

$$g'(z_m) g'(z_p) = \frac{2}{\pi} e^{-\frac{1}{2} \sum_{k,l} (W_{1mk} W_{1ml} + W_{1pk} W_{1pl}) x_k x_l} = \frac{2}{\pi} e^{-\frac{1}{2} \sum_{k,l} A_{kl}^{mp} x_k x_l}$$

i.e., the product of the two derivatives of the activation function result is an unnormalized multivariate Gaussian distribution with inverse covariance matrix $\boldsymbol{A}^{mp}$. The components of this matrix are $A_{kl}^{mp} = W_{1mk} W_{1ml} + W_{1pk} W_{1pl}$.

When taking the expected value, what we will effectively do is calculate the second moments of a multivariate Gaussian distribution, up to a multiplicative constant. This multivariate normal distribution has inverse covariance matrix equal to $\boldsymbol{\Sigma}^{-1} = \boldsymbol{A}^{mp} + \boldsymbol{I}$.

As for the second additive term, noting that $g''(x) = -xg'(x)$ is an odd function, we can directly see that

$$E_x[g''(z_m) x_n x_q] = -\sum_k W_{1mk} E[g'(z_m) x_k x_n x_q] = 0$$

because third moments of a normal distribution are always zero.

**Derivation of the Second Moments**  We want to calculate $E_x[g'(z_m) g'(z_p) x_n x_q]$. We saw that this could be reformulated in terms of the second moments when the distribution is the modified multivariate normal distribution with inverse covariance matrix $\Sigma^{-1} = I + \boldsymbol{A}^{mp}$, where $\boldsymbol{A}^{mp} = W_m^T W_m + W_p^T W_p$ and $W_m, W_p$ are row vectors of the matrix $\boldsymbol{W}_1$. Taking into account the missing prefactor of the normal distribution as well as the factor of $2/\pi$ we have that

$$E_x[g'(z_m) g'(z_p) x_n x_q] = \frac{2}{\pi} \sqrt{\det \Sigma} \int d\boldsymbol{x} \frac{1}{\sqrt{\det 2\pi\Sigma}} x_n x_q e^{-\frac{1}{2} \boldsymbol{x}^T \Sigma^{-1} \boldsymbol{x}} = \frac{2}{\pi \sqrt{\det \Sigma^{-1}}} [\Sigma]_{nq} \ .$$

And so all we have to do is find the determinant of $\Sigma^{-1} = I + A^{mp}$ as well as component $nq$ of its inverse.

To accomplish this we will make use of the following two linear algebra results: the *Sherman–Morrison formula* for the inverse of

$$(A + uv^T)^{-1} = A^{-1} - \frac{A^{-1}uv^T A^{-1}}{1 + v^T A^{-1}u} \ ,$$

valid if $1 + v^T A^{-1}u \neq 0$; and the *matrix determinant lemma* which states that

$$\det(A + uv^T) = (1 + v^T A^{-1}u)\det A \ .$$

By applying the above two formulas inductively we find that the determinant of $\Sigma^{-1}$ is

$$\det(I + A^{mp}) = 1 + W_m W_m^T + W_p W_p^T + W_m W_m^T W_p W_p^T - (W_m W_p^T)^2$$

and that the inverse matrix can be computed as

$$\Sigma = I - \frac{W_m^T W_m}{1 + W_m W_m^T} - \frac{W_p^T W_p}{1 + W_p W_p^T} - \left( \frac{(W_m W_p^T)^2 (\frac{W_m^T W_m}{1 + W_m W_m^T} + \frac{W_p^T W_p}{1 + W_p W_p^T}) - (W_m W_p^T)(W_m^T W_p + W_p^T W_m)}{(1 + W_p W_p^T)(1 + W_m W_m^T) - (W_m W_p^T)^2} \right)$$

**The Lower Right block**   Now, for the $W_2 - W_2$ block, we need to calculate:

$$\frac{\partial^2 \mathcal{L}}{\partial W_{2k} \partial W_{2j}} = E_x[h_k h_j] = E_z[g(z_k)g(z_j)]$$

Because, $z_k = \sum_i W_{1ki} x_i$ and each $x_i \sim N(0,1)$, we have that $z_k \sim N(0, \sum_i W_{1ki}^2)$ and similarly for $z_j$, thus both being marginally normal distributed. However, to take the expected value we need to know their joint distribution. Multivariate statistics tells us that — see for example, Soch et al. (2024) — if $x$ is a multivariate normal distributed variable, $x \sim N(\mu, \Sigma)$, then $z = Ax + b$ is also normally distributed as $z \sim N(A\mu + b, A\Sigma A^T)$.

In our case $\mu = 0$, $\Sigma = I$, where $I$ is the $n \times n$ identity matrix, and $A = \begin{pmatrix} W_k \\ W_j \end{pmatrix}$. Thus, $(z_k, z_j) = z \sim N(0, AA^T)$ where the covariance matrix

$$AA^T = \begin{pmatrix} \sum_{i=1}^n W_{ki}^2 & \sum_{i=1}^n W_{ki} W_{ji} \\ \sum_{i=1}^n W_{ki} W_{ji} & \sum_{i=1}^n W_{ji}^2 \end{pmatrix} \ .$$

So we are now in the position to evaluate

$$E_z [g(z_k)g(z_j)] = \int_{-\infty}^{\infty} dz_j \int_{-\infty}^{\infty} dz_k (2\pi)^{-1} \det(\Sigma)^{-1/2} \operatorname{erf}\left(\frac{z_k}{\sqrt{2}}\right) \operatorname{erf}\left(\frac{z_j}{\sqrt{2}}\right) e^{-\frac{1}{2} z^T \Sigma^{-1} z}$$

**Solving the integral**   For simplicity let $\Sigma^{-1} = \begin{pmatrix} A & F \\ F & B \end{pmatrix}$. We will first integrate one of the error functions using the identity

$$\int_{-\infty}^{\infty} \operatorname{erf}(ax + b) \frac{1}{\sqrt{2\pi\sigma^2}} \exp\left(-\frac{(x-\mu)^2}{2\sigma^2}\right) dx = \operatorname{erf}\left(\frac{a\mu + b}{\sqrt{1 + 2a^2\sigma^2}}\right) \ .$$

And so, keeping only the terms in $z_k$ we want to solve

$$\int_{-\infty}^{\infty} dz_k \operatorname{erf}\left(\frac{z_k}{\sqrt{2}}\right) e^{-\frac{1}{2}\left(Az_k^2 + 2F z_j z_k\right)}$$

Completing the square and using the aforementioned identity, we get

$$\int_{-\infty}^{\infty} dz_k \operatorname{erf}\left(\frac{z_k}{\sqrt{2}}\right) e^{-\frac{1}{2}\left(Az_k^2 + 2F z_j z_k\right)} = e^{\frac{F^2 z_j^2}{2A}} \sqrt{\frac{2\pi}{A}} \operatorname{erf}\left(-\frac{F z_j}{\sqrt{2}\sqrt{A + A^2}}\right)$$

And so, using the previous result, all that is left to do is calculate

$$\int_{-\infty}^{\infty} dz_j (2\pi)^{-\frac{1}{2}} \det(\Sigma)^{-1/2} \operatorname{erf}\left(\frac{z_j}{\sqrt{2}}\right) e^{\frac{F^2}{2A}z_j^2 - \frac{B}{2}z_j^2} \frac{1}{\sqrt{A}} \operatorname{erf}\left(-\frac{F}{\sqrt{A+A^2}}\frac{z_j}{\sqrt{2}}\right) \ .$$

This expression looks rather intimidating. However, if we can find a formula for

$$I(a) = \int_{-\infty}^{\infty} dx \operatorname{erf}(x) \operatorname{erf}(ax) e^{-b^2 x^2} \ ,$$

then we can proceed. To calculate this we will use the Feynman integral trick, i.e., we will differentiate under the integral w.r.t the parameter $a$. We have that

$$\frac{dI}{da} = \int_{-\infty}^{\infty} dx \operatorname{erf}(x) \frac{2}{\sqrt{\pi}} x e^{-a^2 x^2} e^{-b^2 x^2} \ .$$

This integral is solvable and using the software *Mathematica* we find that

$$\frac{dI}{da} = \frac{1}{(a^2+b^2)^{3/2}\sqrt{\frac{1}{a^2+b^2}+1}} \ .$$

Now, to obtain $I(a)$ we just need to integrate both sides of the above from $a' = 0$ to $a' = a$. Noting that $I(0) = 0$, we obtain

$$I(a) = -\frac{2\sqrt{\frac{a^2+b^2}{a^2+b^2+1}}\sqrt{\frac{1}{a^2+b^2}+1}\tan^{-1}\left(-\frac{a\sqrt{a^2+b^2+1}}{b}+\frac{a^2}{b}+b\right)}{\sqrt{\pi}b} + \frac{2\sqrt{\frac{1}{b^2}+1}\sqrt{\frac{b^2}{b^2+1}}\tan^{-1}(b)}{\sqrt{\pi}b}$$

In our case, if we make the substitution $x = \frac{z_j}{\sqrt{2}}$, we can rewrite the integral as

$$\int_{-\infty}^{\infty} dx \sqrt{2}(2\pi)^{-\frac{1}{2}} \det(\Sigma)^{-1/2} \operatorname{erf}(x) e^{-\frac{AB-F^2}{A}x^2} \frac{1}{\sqrt{A}} \operatorname{erf}\left(-\frac{F}{\sqrt{A+A^2}}x\right) \ .$$

Using the previous derivation with $a = -\frac{F}{\sqrt{A+A^2}}$ and $b^2 = \frac{1}{A\det(\Sigma)}$ (which is always positive), we finally obtain, after some simplification steps

$$E_z\left[g(z_k)g(z_j)\right] = \frac{2}{\pi} \arctan\left(\frac{F}{\sqrt{(AB-F^2)(AB+A+B-F^2+1)}}\right)$$

which, when written in terms of the components of the covariance matrix $\Sigma$ is

$$E_z\left[g(z_k)g(z_j)\right] = \frac{2}{\pi} \arctan\left(\frac{\sum_{i=1}^{n} W_{ji}W_{ki}}{\sqrt{1+\sum_{i=1}^{n}W_{ki}^2+\sum_{i=1}^{n}W_{ji}^2+\left(\sum_{i=1}^{n}W_{ki}^2\right)\left(\sum_{i=1}^{n}W_{ji}^2\right)-\left(\sum_{i=1}^{n}W_{ji}W_{ki}\right)^2}}\right)$$

**The Upper Right block**   Here, we need to calculate

$$\frac{\partial^2 \mathcal{L}}{\partial W_{2k}\partial W_{1mn}} = W_{2m}E_x\left[g(z_k)g'(z_m)x_n\right] = W_{2m}\sqrt{\frac{2}{\pi}}\frac{1}{\sqrt{\det \Sigma^{-1}}}E_{\tilde{x}}\left[g(z_k)x_n\right]$$

As we saw previously, $g'(z_m)$ will modify the underlying multivariate normal distribution. Similarly, to the previous calculation, the inverse of the covariance matrix will become $\boldsymbol{\Sigma}^{-1} = \boldsymbol{A}^m + \boldsymbol{I}$, where $\boldsymbol{A}_{kl}^m = W_{1mk}W_{1ml}$ and $\boldsymbol{I}$ is the identity matrix.

Thus, to perform the calculation we need to be able to calculate $E_{\tilde{x}}[g(z_k)x_n]$, where $\tilde{x}$ is the random variable that follows the modified normal distribution. To do the above, we can try to find the distribution of

$\boldsymbol{Y} = (z_k, x_n)$. As $\boldsymbol{x}$ is normally distributed then $y = Mx$ is also normally distributed as $z \sim N(M\mu, M\sigma M^T)$. In this case $\mu = 0, \sigma = (I+A^m)^{-1}$ and $M = \begin{pmatrix} \boldsymbol{W}_k \\ \delta_n \end{pmatrix}$. Thus, $(z_k, x_n)$ follows a normal distribution $N(0, M(I+A^m)^{-1}M^T)$. Using the same techniques as before we find that

$$\boldsymbol{\Sigma} = \begin{pmatrix} W_k W_k^T & W_{kn} \\ W_{kn} & 1 \end{pmatrix} - \frac{1}{1+W_m W_m^T} \begin{pmatrix} (W_m W_k^T)^2 & W_k W_m^T W_{mn} \\ W_m W_k^T W_{mn} & W_{mn}^2 \end{pmatrix}$$

**Calculating the integral** Assuming that we were able to show that $(z_k, x_n) \sim N(\boldsymbol{0}, \boldsymbol{\Sigma})$ with

$$\boldsymbol{\Sigma} = \begin{pmatrix} \sigma_{zz} & \sigma_{zx} \\ \sigma_{zx} & \sigma_{xx} \end{pmatrix} ,$$

we now want to perform the following integral

$$\int_{-\infty}^{\infty} \int_{-\infty}^{\infty} dx_n dz_k \frac{1}{2\pi\sqrt{\det\Sigma}} \exp\left(-\frac{1}{2}\boldsymbol{y}^T \boldsymbol{\Sigma}^{-1} \boldsymbol{y}\right) x_n \operatorname{erf}\left(\frac{z_k}{\sqrt{2}}\right) .$$

With the help of some tabulated integrals of error functions (check Ng & Geller (1969) and Korotkov & Korotkov (2020)) it can be shown that the above yields the following result

$$E[x_n \operatorname{erf}(z_k/\sqrt{2})] = \sqrt{\frac{2}{\pi}} \frac{\sigma_{zx}}{\sqrt{1+\sigma_{zz}}} .$$

Using the previous result we finally have that

$$\frac{\partial^2 \mathcal{L}}{\partial W_{2k} \partial W_{1mn}} = \frac{2}{\pi\sqrt{1+W_{1m}W_{1m}^T}} W_{2m} \frac{W_{1kn} - \frac{W_{1k}W_{1m}^T W_{1mn}}{1+W_{1m}W_{1m}^T}}{\sqrt{1+W_{1k}W_{1k}^T - \frac{(W_{1m}W_{1k}^T)^2}{1+W_{1m}W_{1m}^T}}}$$

# D    Connection between the Hessian and the Fisher information matrix

As stated in the discussion preceding Equation 3, the Hessian we compute at the optimal point is also called the *outer product Hessian* and there is an intimate connection between this Hessian and the Fisher information matrix (FIM). Here, we explicit that connection.

Following Karakida et al. (2021), in the context of regression tasks, the Fisher information matrix is defined as

$$F = \mathbb{E}_{(x,y)} \left[ \nabla_\theta \log p(x, y; \theta) \nabla_\theta \log p(x, y; \theta)^T \right] \tag{19}$$

where the statistical model is given by $p(x, y; \theta) = p(y|x; \theta)q(x)$, where $p(y|x; \theta)$ is the conditional probability distribution of the Neural Network of the output $y$ when given input $x$, that follows some input distribution $q(x)$. The expectation is taken over empirical input-output pairs, $(x, y)$. For regression tasks, the ones considered in this work, the following statistical model is used:

$$p(y|x; \theta) = \frac{1}{\sqrt{2\pi}} \exp \left( -\frac{1}{2} \|y - f(x; \theta)\|^2 \right) . \tag{20}$$

From the definition of the FIM, we have that

$$F_{ij} = \mathbb{E}_{(x,y)} \left[ (y - f_\theta(x))^2 \nabla_{\theta_i} f_\theta(x) \nabla_{\theta_j} f_\theta(x) \right]$$

and so, taking the expectation with respect to $y$, we arrive at

$$F_{ij} = \mathbb{E}_x \left[ \nabla_{\theta_i} f_\theta(x) \nabla_{\theta_j} f_\theta(x) \right]$$

which is exactly the outer product Hessian of the generalization error. At the optimal point, the Hessian just depends on the outer product Hessian and so the spectrum and rank of the Fisher information matrix and the Hessian are the same.

# E Other Choices of Weight Distributions in the Linear Case

If instead of a normal distribution the teacher weights followed a distribution with zero mean and well-defined variance, then the eigenvalues of the first diagonal block of the Hessian, being obtained from the sums of squares of random variables, are subject to the Central Limit Theorem, which ultimately states that their square sum will be governed by some Gaussian distribution. As for the eigenvalues of the second diagonal block of the Hessian, the conditions to arrive at the Marchenko-Pastur distribution are that the distribution for the matrix entries should have zero mean and finite variance (Götze & Tikhomirov, 2004).

The chi-squared distribution with a large number of degrees of freedom behaves as a Gaussian distribution. Thus, when considering another distribution for the teacher weights that has zero mean and, without loss of generality, unit variance, we expect that the Hessian eigenspectrum approximates that of the one obtained from the normal distribution, provided that $N_h$ is large enough. We can see in Figure 8 and Figure 9 examples where this is verified and not verified, respectively.

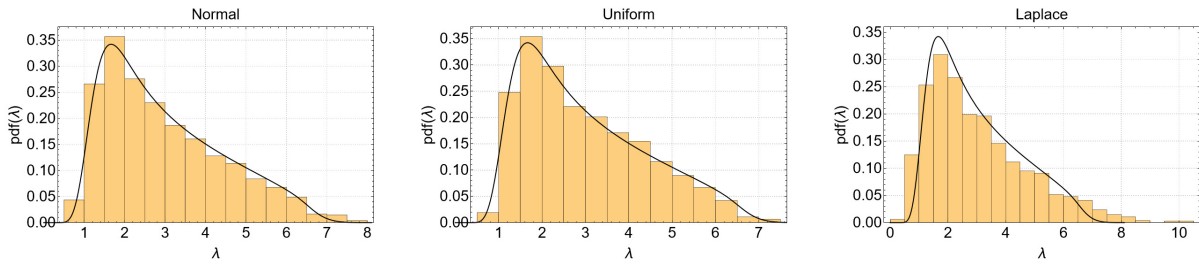

Figure 8: Other choices of distributions for the teacher network weights, showing the agreement for sufficiently large networks. Here, $N_i = 10$ and $N_h = 20$.

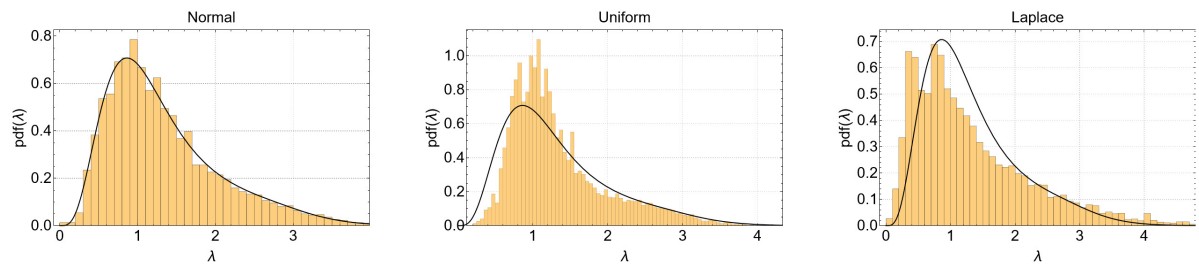

Figure 9: Other choices of distributions for the teacher network weights, showing the differences versus the predicted distribution when the networks dimensions are small. Here, $N_i = 30$ and $N_h = 10$.

