# OpenReview forum: "A Teacher-Student Perspective on the Dynamics of Learning Near the Optimal Point"
_TMLR — Rejected by TMLR_

### Review · Reviewer_UTeW · 2025-02-05

**Summary Of Contributions:**

This paper studies the number of non-zero eigenvalues of the Hessian matrix of MSE loss function in the settings of teacher-student models near the optimal points. With many idealized assumptions, the paper achieved the theoretical results that, if both the student and teacher models are a two-layer linear network, the number of non-zero eigenvalues is at most the input dimension. If both models have polynomial activation functions, upper bounds of the number of non-zero eigenvalues can be theoretically obtained.

**Audience:**

Yes

**Claims And Evidence:**

No

**Requested Changes:**

The paper should build stronger results and remove most of the unrealistic assumptions. Please see the weaknesses above for more details.

**Strengths And Weaknesses:**

There are too many assumptions to achieve the results, many of which are strong and unrealistic.
- A teacher-student setting. Moreover, the teacher and student models are required to have the same architecture.
- weights of the teacher model are required to randomly drawn from a normal distribution, instead of an arbitrary setting mostly seen in literature.
- data are assumed to be distributed according to a normal distribution, which is very strong and unrealistic.
- initialization of weights is required to be close enough to the optimal point, which is not achievable in practice.
- moreover, $\delta \theta(0)=(\theta_0-\theta^*)$ is required to follow a Gaussian distribution. First, this is not doable in practice; second, with this requirement, the results (for example, in the case of linear network, the number of non-zero eigenvalues of the Hessian is no greater than $N_i$) do not generally hold for all parameter settings even near the optimal points. That is to say that the learning dynamics near the optimal points are still mostly unclear, except for a few specific direction with $\delta \theta(0)$ following a Gaussian distribution.
- the activation functions of the neural networks are assumed to be either identity function or polynomial functions. It does not include more commonly used activation functions, such as ReLU, sigmoid, tanh, etc.

While it might be reasonable to assume one or two of the above for simplification of analysis, assuming all is considered too strong.

Even with these assumptions, the results are limited to the region near the optimal point. Moreover, there is no metric to indicate when it is near enough.

Although claiming analyzing the learning dynamics, the paper ends up with one aspect of it (except the simple linear network case), namely the number of non-zero eigenvalues of the Hessian, which is far from describing the whole dynamics.

The results are expected and straightforward. Think about the linear network case: the teacher network is equivalent to a linear model, hence the dataset is essentially generated from a linear model. It is also known that the outer product Hessian essentially has the same spectrum as NTK (except some zero-eigenvalues), and that NTK matrix of linear model is just $XX^T$, the rank of which is at most $N_i$. Therefore, the (outer product) Hessian has at most $N_i$ positive eigenvalues. Similar analysis can be applied to quadratic and polynomial networks, and the rank of their NTKs can be easily analyzed to get the result of the paper.

The claim of the paper that "the eigenspectrum ... exhibits a bulk of eigenvalues near zero, with a single eigenvalue being much farther apart from the bulk" is not theoretically verified.

---

> ### Author Response · Authors · 2025-04-15
> **Response to Reviewer UTeW (Part 1)**
>
> We thank the reviewer for the detailed feedback. Regarding the several assumptions made to achieve our results, we believe most follow what is common in the literature. Nevertheless, we have some specific comments to add about each assumption:
> - "A teacher-student setting with the same architecture": This setting is indeed a strong assumption; however, to study the dynamics near an optimal point we cannot forego it. As mentioned in the conclusions, we could try to overparameterize the student network to relax this assumption, but still, the optimal solution would, at most, vary by linear combinations of the student weights W2 to achieve the same W2 vector in the teacher network. As mentioned in the paper, we leave such a study for future work, focusing only on the case with the same architecture, which is an assumption also made by previous literature [3].
> - "Weights of the teacher model are required to be randomly drawn from a normal distribution": Everything up to the determination of the eigenspectrum of the Hessian does not depend on the choice of the distribution of teacher weights. For example, in the linear case, it still is true that, for most choices of arbitrary teacher weights, the positive eigenvalues follow the reasoning in Appendix A.3. So, our observations on the Hessian rank do not depend strongly on this. The assumption of an underlying distribution for the teacher weights is used in order to describe a distribution for the eigenvalues of the Hessian and is not new in the literature [2, 4]. Furthermore, for most distributions for the teacher weights, with zero mean and well-defined variance, and for large enough networks, we obtain the same eigenvalue distribution as the one in the main text. We added Appendix E showing this numerically for other choices of weight distributions for the teacher.
> - "Data are assumed to be distributed according to a normal distribution": This is a common assumption in the literature [1, 2, 3, 4]. Here, it is necessary to simplify the analysis of Hessian components for the generalization error. We have verified numerically that choosing another input data distribution, like the uniform distribution, influences the Hessian components but does not affect the Hessian rank.
> - "Initialization of weights is required to be close enough to the optimal point": By "near the optimal" point, we mean that the student network is already in a locally convex region around the minimum. Even in practical cases, for small enough learning rates after enough training we would be in such case, albeit potentially around a local minimum. In this work we focus on optimal point, but the analysis done here for the linear case remains correct, provided we compute the Hessian eigenspectrum around the local minimum we arrived at.
> - "$\delta \theta (0) = (\theta_{0} - \theta^*)$ is required to follow a Gaussian distribution.": This is a technical assumption that simplifies the analysis. The main result here, that the long-time learning dynamics is dominated by the smallest non-zero eigenvalue, remains valid, as long as the corresponding eigenvector does not change abruptly near the optimal point.
> - "The activation functions of the neural networks are assumed to be either identity function or polynomial functions.": Linear and quadratic activation functions were chosen based on having analytical solutions for the corresponding Gaussian integrals for the Hessian components. Still, we study a non-polynomial function that has a sigmoid shape, the error function (erf). While not as popular as other choices of activation functions, we believe it is still relevant.
>
> #### References
>
> [1] - Goldt, Sebastian, Madhu S Advani, Andrew M Saxe, Florent Krzakala, and Lenka Zdeborová. 2020. ‘Dynamics of Stochastic Gradient Descent for Two-Layer Neural Networks in the Teacher–Student Setup’. Journal of Statistical Mechanics: Theory and Experiment 2020 (12): 124010. https://doi.org/10.1088/1742-5468/abc61e.
>
> [2] - Pennington, Jeffrey, and Pratik Worah. ‘The Spectrum of the Fisher Information Matrix of a Single-Hidden-Layer Neural Network’. In Advances in Neural Information Processing Systems, Vol. 31. Curran Associates, Inc., 2018. https://papers.nips.cc/paper_files/paper/2018/hash/18bb68e2b38e4a8ce7cf4f6b2625768c-Abstract.html.
>
> [3] - Arjevani, Yossi, and Michael Field. 2020. ‘Analytic Characterization of the Hessian in Shallow ReLU Models: A Tale of Symmetry’. In Advances in Neural Information Processing Systems, 33:5441–52. Curran Associates, Inc. https://proceedings.neurips.cc/paper/2020/hash/3a61ed715ee66c48bacf237fa7bb5289-Abstract.html.
>
> [4] - Karakida, Ryo, Shotaro Akaho, and Shun-ichi Amari. 2021. ‘Pathological Spectra of the Fisher Information Metric and Its Variants in Deep Neural Networks’. Neural Computation 33 (8): 2274–2307. https://doi.org/10.1162/neco_a_01411.

---

> ### Author Response · Authors · 2025-04-15
> **Response to Reviewer UTeW (Part 2)**
>
> Regarding the dynamics near the optimal point for non-linear activations, the issue is that the bulk of positive eigenvalues near zero dominate the behaviour. As shown in the paper, the loss dynamics near the optimal point depend on both the eigenvalues as well as the corresponding eigenvectors. Because most eigenvalues are highly concentrated near zero the analysis is more complicated than in the linear case, where we could focus on the smallest positive eigenvalue.
>
> We thank the reviewer for pointing out the connection between the Neural Tangent Kernel (NTK) and the Hessian. While it is true that the rank results can be derived via the NTK, due to the connection between the positive spectrum of the Hessian and the NTK, the remainder of the results are easier to obtain from the Hessian perspective.
>
> Finally, regarding the claim that "the eigenspectrum ... exhibits a bulk of eigenvalues near zero, with a single eigenvalue being much farther apart from the bulk", this is an observation taken from the numerical diagonalization of the Hessian. It is not a theoretical claim, because for the quadratic and erf activation functions, we were not able to theoretically describe the spectrum like we did for the linear case. We have clarified this in the revised manuscript.

---

### Review · Reviewer_zDQg · 2025-02-09

**Summary Of Contributions:**

This work studies the Hessian spectrum of nonlinear networks

**Audience:**

Yes

**Claims And Evidence:**

Yes

**Requested Changes:**

Due to the weakness I point out above, I believe that the authors should thoroughly compare the Hessian spectrum and the Fisher spectrum, and find a case to argue that Hessian is a better metric of the effective dimension than the Fisher

**Strengths And Weaknesses:**

Strength:
(1) the hessian of neural networks is an interesting and fundamental problem

Weakness:
I just feel that the contribution is too weak, and that the finding is accidental

There is a much better known and widely accepted metric of effective number of parameters: the rank of the Fisher information. And, on top of that, it is very often the case that the Fisher information matrix approximates the Hessian. Therefore, the finding that Hessian determines the effective dimension may be nothing but a reflection that Hessian approximates the Fisher information, which actually determines the effective dimension.

---

> ### Author Response · Authors · 2025-04-15
> **Response to Reviewer zDQg**
>
> We thank the reviewer for the provided feedback and for pointing us towards the connection between our work and the Fisher information matrix (FIM). Previous works [1, 2] study spectral properties of the FIM and we revised the section on related work to take this into account (see also below).
>
> In [1], the equality between the empirical FIM and the empirical Hessian at a global minimum is shown. Based on it, we added Appendix D to the manuscript, detailing the connection between the Hessian and the Fisher information matrix. As shown there, for regression tasks, the Hessian matrix for the generalization error at the optimal point and the Fisher information matrix coincide. Consequently, all results regarding the rank and eigenspectrum of the Hessian are valid for the Fisher information matrix.
> In [2], a limit is considered, with the input, hidden and output dimensions all assumed to be equal and taken to infinity. As can been seen in Theorem 1 there, the spectrum of the FIM is derived and depends only on the activation function. On the contrary, we derive exact expressions for the Hessian components for the generalization error for any $N_{i}$ and $N_{h}$, only assuming a real output.
>
> As for the final request, under the assumptions of our work, at the optimal point, the Hessian and the FIM are equal, as mentioned above. Thus, we cannot, in our work, fundamentally distinguish between them or argue one over the other. Notwithstanding, we agree with the reviewer that it is an interesting question to study the difference between the Hessian rank and the FIM rank, away from the optimal point, to assess the better metric for the effective parameters.
>
> [1] - Karakida, Ryo, Shotaro Akaho, and Shun-ichi Amari. 2021. ‘Pathological Spectra of the Fisher Information Metric and Its Variants in Deep Neural Networks’. Neural Computation 33 (8): 2274–2307. https://doi.org/10.1162/neco_a_01411.
>
> [2] - Pennington, Jeffrey, and Pratik Worah. ‘The Spectrum of the Fisher Information Matrix of a Single-Hidden-Layer Neural Network’. In Advances in Neural Information Processing Systems, Vol. 31. Curran Associates, Inc., 2018. https://papers.nips.cc/paper_files/paper/2018/hash/18bb68e2b38e4a8ce7cf4f6b2625768c-Abstract.html.

---

### Review · Reviewer_3G3o · 2025-04-01

**Summary Of Contributions:**

The paper considers a teacher-student setup, with matched architectures (1-hidden-layer), for linear, quadratic and networks with error function activations, and explores the Hessian at the weights where the student also matches the weights of the teacher. At this configuration, the eigenspectrum of the Hessian of the student with respect to data is derived -- and its rank is proposed as a measure of the effective number of parameters of the network.

**Audience:**

Yes

**Broader Impact Concerns:**

No concerns.

**Claims And Evidence:**

Yes

**Requested Changes:**

In reading this work, especially the introduction and motivation, I had some difficulty following the reasoning for the analysis in this work. Since it may be a big misunderstanding, I will lay out the main points I find concerning here, so that they may be adressed.

**weaknesses**

1. *Initialization near the optimal point is unclear.* In the analysis the author consider the second derivative of the teacher at its weights and analyze how gradient flow would behave around it, in the sense that it looks at the curvature in terms of the Hessian. It is unclear what the authors mean by near - and how near one would have to be in practice to find this behaviour. Since initializations of training dynamics are typically random and the landscape of complex real world networks includes many minima this question is relevant. Of course this is not necessarily the case for 1-layer linear networks, but even here symmetries and different scalings lead to many other solutions. How would the analysis of the "optimal point" would relate to learning dynamics that end up near other optimal points?

2. *The teacher-student naming is misleading.* In the current literature, as also cited by the authors e.g. Goldt et al (2020), the teacher student setup is usually understood as the student learning from a set of datapoints that are generated by the teacher, finite, online or in batches. The interesting part is understanding how the training and test (generalization) error are related in this case, as many versions of the student (due to symmetries) might approximate the teacher perfectly in terms of loss, but finding them algorithmically is not a given. By assuming that the student just copies the correct parameters from the teacher, the effects from symmetries and the learning dynamics from a random initialization are completely excluded from the type of analysis that is made here.
 In addition, by approximating the sum over training data in the emprical risk with the integral over the Gaussian distribution, one essentially switches to analzing the generalization error - so the analysis here is not even a static analysis of an empricial risk landscape, but rather under the hypothesis that the learner knows the full distribution. In this case, I would find a naming such as "analysis of the Hessian of a random neural network around its parameters in the generalization loss" less misleading. If there is one, could the authors point out my misunderstanding and motivate more clearly why they ended up introducing the problem as it is?

3. *However, the link of the properties of the Hessian with the training dynamics has, to our knowledge, been overlooked.* I find this statement very strong and I do not agree. First of all, there are methods for training neural networks that consider second order properties as the Hessian, then, there has been previous work that analyses this even for deep nets, e.g.https://arxiv.org/abs/1811.07062 or https://arxiv.org/abs/1907.10732 Could the authors point out more clearly where exactly their qualitative and technical contributions lie with respect to existing work?

**strengths**

Gerneally I think the idea of understanding the properties of the weights at the teacher in the generalization loss could be interesting, considering e.g. how optimal algorithms would behave around it - but as it is written in the paper now I do not see this clearly.

Some nitpicks to consider that I think would improve the work:
- The authors consider a regression case, i.e. a single neuron output, it would be good if this was mentioned explicitly in e.g. the introduction
- $\hat y$ is usally for the estimator (the student) - later on $\theta^*$ is used for the teacher more in line with related work, it would be useful to adhere to a single notation.
- I have a hard time understanding why the plots in e.g. Figure 4 contain empy section in the lines. In any case, it would also be could to see averages and standard deviations of several runs.

**Strengths And Weaknesses:**

See below, I think I do not have a clear understanding of the objectives, which is why I summarize them in "requested changes".

---

> ### Author Response · Authors · 2025-04-15
> **Response to Reviewer 3G3o**
>
> We thank the reviewer for the detailed comments. The main objectives of this work are to study the optimal point in a teacher-student setting and to understand both the near-optimal dynamics and the role that the Hessian rank plays in such an idealized setting when considering different choices for the activation function.
>
> We now move on to address your concerns:
> - Regarding the first point, when we say "near the optimal point", we mean that the student network is already in a locally convex region of the loss landscape. Even though our work focuses on the optimal solution with matching weights, the analysis of the loss dynamics near local minima and other optimal solutions would yield the same results, provided we know the eigenspectrum of the Hessian at those minima. We can comment further on the other optimal solutions that arise due to discrete symmetries like permutations of neurons because these do not alter the spectrum of the Hessian and, as such, do not change our results.
> - As for the second point, here we respectfully disagree. Although we only study the generalization error and not the connection between it and the empirical risk, we still base our entire analysis on having a (random) teacher-student setup. Not only do we use the setup to know the optimal solution, but also numerically train the student with data generated by the teacher network when studying the dynamics near the optimum, as seen in the loss curve in Figure 3. We believe that moving away from the "teacher-student" term might lead to further confusion regarding the base assumptions of our work. The main objective is to study the Hessian of the generalization error precisely when we are in a teacher-student setup, with the student at the optimal solution.
> - Addressing the third point, we agree with the reviewer that our previous statement was too strong, and so we retracted it.
>
> Finally, regarding the nitpicks, thank you for the suggestions. We have revised the manuscript: explicitly mentioning that we consider a regression task; clarifying the notation for the optimal parameters and teacher output; and explaining the missing lines in Figure 4, which represent gaps between the bulks of observed eigenvalues where no eigenvalues were numerically found.

---

### Decision · Action_Editor_gWeD · 2025-08-19

**Recommendation:** Reject

**Audience:**

No

**Audience Explanation:**

As pointed by all reviewers, the contribution is rather modest. 2 recommend a rejection based on this. Novelty is not an acceptance criteria for TMLR. However, the potential interest of some of its readers is, which I believe is lacking for the following reasons. The setting considered is the idealized teacher-student setting, which lies at the core of a huge body of literature in statistical physics and beyond, at least since the eighties. The counterpart of idealizing so much the setting (target neural network teacher, student with matched architecture, gaussian data etc) is that one can generally obtain very precise and quantitative statements. However, from my own additional reading, the precise statements in the paper are very restricted (linear network). Beyond the linear network case, some calculations on the Hessian near the optimal point can be performed which do not bring much insights. These calculation yield spectra of Hessian (based on numerical evaluations) but not much discussion/conclusion is provided, in particular with respect to the dynamical aspects that the paper claims at analyzing, while many strong results on the dynamics in this precise model are known (and no discussion/connection is established with these). This is an issue. I therefore do not believe that the paper will grasp the interest of this statistical physics community. The derivations are exploiting some random matrix theory tools, but rather simple results on that side. There exists by now a very developed theory of the random matrices appearing in the context of machine learning, based on more advanced concepts. I do not foresee enough interest either from the community working on the random matrix theory applied to learning either. Given that one of the criteria for acceptance in TMLR is related to the potential interest by at least some part of the TMLR audience, I recommend rejection. I believe the authors should re-submit the paper after having discussed in greater details the implications of their results, and putting these in perspective with the related literature, in particular classical result on the dynamics (e.g. which exploit statistical physics techniques).

**Claims And Evidence:**

Yes

**Claims Explanation:**

The paper supports its claims based on detailed and sound computations, as two reviewers agree with, which I confirm from my own reading. The theoretical results are supported by easy-to-reproduce simulations.

**Resubmission Of Major Revision:**

The authors may consider submitting a major revision at a later time.